# Increased Risk of Alzheimer’s Disease in Patients with Head and Neck Cancer

**DOI:** 10.3390/cancers15235516

**Published:** 2023-11-22

**Authors:** Il Hwan Lee, Hyunjae Yu, Seung-Su Ha, Hee Gyu Yang, Dong-Kyu Kim

**Affiliations:** 1Department of Otorhinolaryngology-Head and Neck Surgery, Chuncheon Sacred Heart Hospital, Hallym University College of Medicine, Chuncheon 24253, Republic of Korea; 2Institute of New Frontier Research, Division of Big Data and Artificial Intelligence, Chuncheon Sacred Heart Hospital, Hallym University College of Medicine, Chuncheon 24253, Republic of Korea

**Keywords:** cancer, head, neck, Alzheimer’s, cohort

## Abstract

**Simple Summary:**

Several studies have reported that cancer could be associated with developing Alzheimer’s disease (AD). However, whether head and neck cancer (HNC) causes an increased risk of developing AD still remains unclear. Thus, we aimed to evaluate the correlation between HNC and AD. We found that HNC, in particular oral cavity cancer, had an increased incidence of AD events. Additionally, they occurred more frequently in females, middle-aged patients, and within the first year after HNC diagnosis. Our findings suggest that clinicians should closely evaluate the development of AD when they treat HNC patients.

**Abstract:**

Patients with head and neck cancer (HNC) often experience cognitive impairment. However, the relationship between cancer and Alzheimer’s disease (AD) remains unclear. We aimed to elucidate the relationship between patients with HNC and their subsequent AD development. This retrospective study used data from a nationwide representative cohort sample, the Korean National Health Insurance Service Cohort. The cancer group was defined based on the presence of diagnostic codes for HNC (C00-C14 and C30-C32). After matching the independent variables with a propensity score of 4:1, a total of 2304 people without HNC and 576 with HNC were enrolled in this study. Hazard ratios (HRs) of AD incidence (per 1000 person-years) and 95% confidence intervals (CIs) in HNC patients were calculated. The incidence of AD was 14.92 in HNC patients and 9.77 in non-cancer patients. Additionally, the HNC group was found to have a higher risk of developing AD compared with the non-cancer group. Female and middle-aged HNC patients had a higher risk of developing AD events compared with other subgroups. Surprisingly, during the observation period, the risk of developing AD was relatively high within the first year after HNC diagnosis. In conclusion, our study suggests that HNC and AD are positively correlated.

## 1. Introduction

Head and neck cancer (HNC) collectively refers to malignant tumors that occur in the head and neck region, including the mouth, throat, larynx, nose, sinuses, and neck. [1]. Major risk factors for developing HNC include drinking, smoking, eating habits, and geographic characteristics. In South Korea, the incidence rate of HNC is 5–6 per 100,000 person-years, which is lower than that of other sites; however, HNC shows a high mortality rate of 1.5 per 100,000 person-years [2]. Advances in cancer diagnostic technology and improvements in essential treatments for patients with HNC, such as surgery, radiation therapy, and chemotherapy, have enabled patients with cancer to survive longer, which has also increased the rate of late complications. Long-term side effects such as congestive heart failure, premature menopause, and cognitive problems, such as Alzheimer’s disease, result from the cancer or its treatment, which can occur months or years after treatment. Among these, cognitive decline is closely related to a decrease in quality of life and can place a great burden on patients and their families, causing problems in local communities [3]. This decline in cognitive function and quality of life may also affect treatment compliance and outcomes.

Typically, we recognize dementia when acquired cognitive impairment becomes severe enough to impair social or occupational functioning. Alzheimer’s disease is one of the most common diseases that cause these dementia symptoms. It is a progressive disease that initially begins with mild memory loss; however, it gradually worsens, eventually leading to loss of ability to communicate and maintain daily activities. In addition, Alzheimer’s disease is associated with structural and functional problems in the part of the brain that controls thinking, memory, and language. The mechanisms underlying cognitive impairments are multifactorial. Although definitive evidence has not yet been established, several previous studies report conflicting views on the association between cancer and cognitive impairment [4,5,6]. Some studies have reported that dementia occurred less in patients with cancer than in normal controls, and that cancer and dementia showed an inverse association with each other [4,7]. However, there is a possibility that surveillance bias and survival bias may have existed in these studies; hence, they should be cautiously interpreted [6]. On the contrary, some studies reported that the risk of developing dementia is increased in cancers such as prostate and gastric cancer; moreover, cancer treatment methods have different effects on the development of dementia [8,9]. The association between various malignancies and dementia is well reported in previous studies. However, no study to date has reported a correlation between HNC and dementia, especially Alzheimer’s disease.

Therefore, we examined the incidence and potential risk of Alzheimer’s disease in patients with HNC using a nationally representative cohort dataset. This dataset includes all types of diseases based on the nationwide population; thus, we identified a possible link between two specific diseases. We also controlled for suspected confounders, such as clinical status and demographic conditions.

## 2. Materials and Methods

### 2.1. Ethical Considerations

This study used a nationwide population-based cohort dataset provided through a de-identification process based on health insurance claims data collected by the National Health Insurance Corporation. This cohort dataset has a relatively large sample size and very low attrition rate over 10 years. Immigrants are excluded from this cohort. However, representativeness is not compromised, because South Korea has a very low proportion of immigrants from abroad. The healthcare provider database also included facility types, personnel, and equipment information. The database contains a wide range of medical utilization information, including dates of death, visits to hospitals and outpatient medical facilities, and medication history. However, the status of disease diagnoses in claims data may not accurately reflect the patient’s health status. Thus, the operative definition of diagnosis was needed to enhance the accuracy. The Institutional Review Board (IRB) of Hallym Medical University, Chuncheon Sacred Hospital, approved this study (IRB No. 2021-08-006), and waived the need for written informed consent due to the de-identification process. Because the data are de-identified, the process of obtaining consent through subject consent and direct contact was exempted. However, the original dataset cannot be released to the public due to government policy.

### 2.2. Cohort Dataset and Study Design

The cohort dataset used in this study contains key medical information, including hospitalization and outpatient visits, procedures, and prescriptions, for a total of 1,025,340 adults. If the diagnosis code is entered in accordance with the International Classification of Diseases, 10th Revision, Clinical Modification (ICD-10-CM) standard, there is a unique identification number assigned at birth, so all types of medical claim data registered in the cohort dataset would not be overlapped or be omitted. Therefore, a nationally representative cohort dataset can reflect the entire population of South Korea while minimizing selection bias. Additionally, this cohort data set has proven to have an excellent reliability rating through a previously conducted reliability verification study [10,11]. Several previous studies examining the relative association between two specific diseases have used the same cohort dataset as ours [12,13,14,15].

Because this study was designed using a retrospective cohort approach, all data are obtained from records. Thus, in this longitudinal study, we started observations from a specific time point in the past, and detected the results that occurred between that time point and the present. Even though the results occurred in the past, we could compare groups of individuals who are similar in many ways, but who differ on certain characteristics in terms of specific outcomes. As a washout period, we eliminated the first year (January to December 2002) of the cohort dataset; thus, we could eliminate the presence of dementia events before the diagnosis of HNC. We enrolled patients with HNC during the index period (2003–2005). The cancer group was defined based on the presence of diagnostic codes for HNC (oral cavity cancer: C00-C06, salivary cancer: C07-C08, oropharynx cancer: C09-C10, nasopharynx cancer: C11-C12, hypopharynx cancer: C13-C14, sinonasal cancer: C30-C31, and larynx cancer: C32). We also selected individuals who experienced these diagnostic codes more than twice within the index period, or were hospitalized with these diagnostic codes. Additionally, to increase the accuracy of the analysis, the exclusion criteria were set as patients under 55 years of age, patients who died during the index period, and patients diagnosed with dementia before cancer diagnosis. A propensity score-matching method was then used to enroll non-cancer participants as a comparison group. For each cancer patient, four matched cancer-free participants were selected. At this time, all independent variables and year of registration (cancer diagnosis) were matched between each matched subject in the comparison group and cancer group. The primary end point in the study was defined as the cohort date on which a specific event (Alzheimer’s disease [F00, G30]) occurred, and if the specific event did not occur in a patient until the final follow-up period in the database, the patient was censored at the end point of the database. Figure 1 outlines the study design and study participant registration process.

### 2.3. Independent Variables

In the present study, we controlled for several independent variables between the cancer and non-cancer cohorts to elaborate on the outcomes. Thus, we chose several independent variables such as age, sex, residence, household income, and comorbidities. The eligible database included information about income-based insurance contributions (a proxy for income). For comorbidity matching, a weighting index, the Charlson Comorbidity Index (CCI), was used. This is a method commonly used in claims data research. It assigns different weights to specific conditions and adds the weights to find an index for a specific patient. Using CCI, we matched participants’ comorbidity levels by dividing them into 0, 1, and 2.

### 2.4. Statistical Analysis

The incidence of Alzheimer’s disease is calculated by measuring how often a disease or other event occurs during a specific period and is expressed in units of 1000 person-years. In this study, person-years for incidence assessment were calculated using the following method. First, when a registered patient died, the number of years from the first diagnosis of HNC to the date of death was calculated. Second, cases of developing Alzheimer’s disease were calculated as the number of years from the first diagnosis of HNC to the first diagnosis of Alzheimer’s disease. Lastly, if no events occurred, the number of years from the first diagnosis of HNC until the end of the study was calculated. The risk of Alzheimer’s disease in HNC patients was assessed using weighted Cox regression. These results were expressed as crude and weighted hazard ratios (HRs) with 95% confidence intervals (CIs). For all statistical analyses in this study, R software (version 3.5.0, R Foundation for Statistical Computing, Vienna, Austria) was used. Statistical significance was applied based on *p* = 0.05.

## 3. Results

### 3.1. General Characteristics of the Cohort Population

After matching the independent variables set in the study design, a total of 2304 non-cancer patients and 576 cancer patients were selected for analysis in this study. For the HNC and non-cancer groups, all covariates used for sample matching were similarly distributed between the two groups. We found no significant differences in each independent variable between the two cohorts (Table 1). Finally, we performed a balance plot test to ensure that appropriate matching was achieved. When determining the above analysis comprehensively, we were able to confirm that the two cohort groups were appropriately matched.

### 3.2. Incidence Rate of Alzheimer’s Disease

To calculate incidence rates, person-years were assessed in both groups (19,249.3 person-years for the non-cancer group, and 3953.7 person-years for the HNC group). Interestingly, we observed a higher incidence of AD in the HNC group (14.92 per 1000 person-years in the cancer group and 9.77 per 1000 person-years in the non-cancer group) (Table 2).

### 3.3. Risk Rate of Alzheimer’s Disease

Univariate and multivariate weighted Cox regression models were used to evaluate the risk ratio of the incident AD events during the follow-up period (Table 3). We observed a significantly increased risk of developing incident AD events in HNC patients compared with non-cancer patients after adjusting for all covariates (weighted HR = 1.65, 95% CI = 1.23–2.22). Moreover, when the risk during the follow-up period was evaluated, the risk of Alzheimer’s disease was found to be relatively high in the first year after HNC diagnosis. Meanwhile, although the risk decreased after the first year of the HNC diagnosis, a significant risk ratio of developing Alzheimer’s disease remained throughout the follow-up period (Figure 2).

### 3.4. Subgroup Analysis

Subgroup analyses were performed to determine the influence of gender, age, and HNC subtype. When analyzed by sex, female patients with HNC were found to have a significantly increased risk of developing Alzheimer’s disease, unlike male patients (Table 3). When analyzed by age, we observed an increased risk of subsequent AD in patients with HNC in both age categories; however, middle-aged patients had a higher risk compared to older people (Table 4). Lastly, to determine the risk of Alzheimer’s disease according to the HNC subtype, the cancer group was divided into the oral cavity, salivary gland, oropharynx, nasopharynx, hypopharynx, sinus tract, and larynx group, based on the cancer site (Table 5). Results from Cox analysis showed that the weighted risk ratio for the incident AD events was significantly higher in the oral group at 1.74 (95% CI: 1.27–2.83). However, other subtypes did not show a significant positive correlation between the two diseases.

## 4. Discussion

Worldwide, HNC is the sixth most common cancer type, and is one of the malignancies whose long-term survival remains low despite recent advances in treatment methods. Among the complications associated with HNC, cognitive impairment is common, and is closely related to decreased quality of life and psychosocial problems. Recently, research on Alzheimer’s disease and related diseases is increasing, in particular, research on the correlation between Alzheimer’s disease and cancer. [4,5,6,16]. Several previous studies have analyzed the correlation between Alzheimer’s disease and various types of malignant tumors, including colon cancer, stomach cancer, prostate cancer, and nasopharyngeal cancer, and these results have conflicting conclusions, showing both inverse and positive associations. [5,8,9,17].

This study is the first to investigate the incidence and risk ratio of Alzheimer’s disease in patients with HNC using a longitudinal population-based cohort dataset. We found that the incidence of Alzheimer’s disease in HNC patients was higher than that in non-cancer patients. Additionally, when adjusting for key independent variables, we observed that the risk ratio for Alzheimer’s disease in the HNC group was significantly higher than that in the non-cancer group (adjusted HR = 1.65, 95% CI = 1.23 to 2.22). The HNC’s most frequent tumor sites are the larynx, oral cavity, and pharynx. HNCs also include salivary gland tumors, nasopharyngeal cancer, and paranasal and nasal sinus cancer. In the subtype analysis of HNC, oral cavity cancer showed a significantly higher risk of incidence of Alzheimer’s disease (adjusted HR = 1.74, 95% CI = 1.27–2.83). However, besides oral cavity cancer, the enrolled patients with other cancer types were too small in the analysis (Table 5). Thus, we could not conclude that only oral cavity cancer had a significant association with Alzheimer’s disease.

The possible mechanism of action of HNC in the development of Alzheimer’s has not yet been clarified. However, the evidence of an association between these two diseases can be inferred from several previous studies. First, treatment methods for HNC can also affect the development of Alzheimer’s disease. The management of HNC is often a clinical challenge, since the disease is locally advanced at diagnosis in more than 60% of patients. The main treatments for HNC are divided into surgery alone, radiation therapy after surgery, and radiation therapy alone, which is performed in almost all patients to increase the survival rate [18]. Generally, when HNC is deemed unresectable or an organ preservation goal is pursued, the current standard treatment is a combination of radiation and chemotherapy. In particular, oral cavity cancer, a subtype that showed significantly higher HR among HNC in our study, is likely to cause radiation-related brain damage because the treatment site is adjacent to the brain when radiation therapy is administered. One meta-analysis analyzed seven studies and reported that patients with HNC who underwent radiation therapy showed cognitive decline, and concluded that radiation therapy reduced neurocognitive function by causing microstructural and functional changes in the brain [19]. Additionally, one previous cohort study showed that patients with nasopharyngeal cancer treated with radiation therapy were more likely to develop dementia than normal controls [5]. In a subtype analysis of dementia, Alzheimer’s disease showed the highest HR compared to other types of dementia. Another cohort study [20] reported the risk of dementia to be radiation dose-dependent, particularly in patients under 65 years of age; the risk of dementia was 3.54 times higher in patients who underwent radiotherapy alone than in patients who underwent surgery alone. Some studies have focused on brain volume, in areas such as the hippocampus—which is associated with memory, learning, and emotional functions—and its relationship with radiation therapy [21,22]. A resonance imaging study demonstrated that patients with nasopharyngeal cancer treated with radiation therapy had a significantly greater volume reduction in the bilateral hippocampus than normal controls [22]. Vascular injury and temporal lobe necrosis due to radiation are cited as the biggest causes of radiation therapy affecting the risk of developing dementia [23,24]. Radiation therapy induces atherosclerotic progression to blood vessels and increases occlusive changes, resulting in radiation-related ischemia [25,26]. Temporal lobe necrosis is a late-stage complication that occurs after radiation therapy and, if aggravated, is accompanied by headaches, dizziness, and cognitive impairment, which can be a precursor to dementia [24]. In addition, an in vitro experimental study to investigate the effect of radiation on the regeneration of neural cells reported that radiation arrested the cell cycle of neural stem cells and impacted cellular differentiation [27]. Based on these previous studies, it can be considered that radiation therapy may cause damage to the temporal lobe and interfere with neural regeneration, thereby affecting cognitive function.

Additionally, complications associated with HNC surgery may affect the development of Alzheimer’s disease. Compared with other cancers, HNC causes many inconveniences in daily life, because it is accompanied by cosmetic problems and a serious decline in the ability to speak or eat. Therefore, more patients with HNC experience anxiety and depression than patients with other cancer types, which are more severe in women [28,29]. Several studies have reported that depression has a positive association with the development of Alzheimer’s disease [30,31,32,33]. One study showed that recent depression was independently associated with an increased risk of Alzheimer’s disease, and reported that women had a higher risk than men [30]. Some studies have shown that depression accelerates brain aging and reduces dehydroepiandrosterone, which protects neurons against N-methyl-D-aspartate-induced neurotoxicity while increasing cortisol secretion, thus increasing the risk of Alzheimer’s disease [34,35,36]. In our study, female patients with HNC showed a higher risk of Alzheimer’s disease than male patients, possibly because women are more sensitive to psychological impacts after surgery.

Moreover, common environmental exposures, such as smoking, can increase the association between HNC and Alzheimer’s disease. In our study of HNC subtypes, cancers of the oral cavity, oropharynx, and larynx, which are at high risk for Alzheimer’s disease, are all significantly affected by smoking status, and smoking is a well-known strong risk factor for HNC [37]. Several studies also have reported that smoking poses a high risk for the development of Alzheimer’s disease, which was found to be related to the smoking period and amount of smoking [38,39,40,41]. Although further studies are needed to determine exact pathophysiological mechanisms, the various common pathophysiologies described above indicate the existence of a close correlation between HNC and Alzheimer’s disease. Finally, there are reports that a genetic correlation exists between cancer and Alzheimer’s disease. A previous genome-wide statistics study has demonstrated a considerable genetic overlap between cancer and dementia [42]. In particular, the strongest positive genetic correlations were found in regions of the genome displaying enhancer signatures, indicating that the regulation of gene expression may play a common role in the pathogenesis of both diseases [6].

The unique strengths of this study are as follows. First, this study is the first cohort study to evaluate the incidence and risk of Alzheimer’s disease in Korean HNC patients using nationwide population-based data. Health insurance claims cohort data databases are built using claims data accumulated while the health insurance system operates. Therefore, they can provide information related to health promotion, prevention, diagnosis, treatment of diseases and injuries, rehabilitation, childbirth, and death. Thus, we applied matching of sexual behavior scores to see if we could control for important confounding facts that determine whether the two groups have problem-solving issues. This means that the incidence rates in this study are found to be very comparable. However, due to the limitations of the retrospective cohort study design, it was not possible to accurately determine whether the associations found in our analysis were coincidental or new findings between the two groups. However, our analysis results have very important clinical significance when treating patients in actual clinical settings. Secondly, this study includes a relatively large number of patients and a long follow-up period (10 years). Because the cohort database contains extensive information related to visits to medical institutions, including both outpatient and inpatient visits, a large number of patients and a long-term follow-up period are essential for increasing analysis power. Finally, this nationwide cohort data are based on claims data and include sociodemographic data, healthcare utilization, health screening data, and healthcare provider information. Thus, we adjusted for all possible sociodemographic characteristics. These advantages allow us to minimize errors due to surveillance bias in assessing the risk of AD in HNC patients.

However, this study has some limitations that require careful interpretation. First, this study identified diseases based on diagnosis codes included in the cohort dataset, and did not identify diseases by considering medical history or pathological findings. Therefore, it was not possible to conduct an analysis according to disease severity and affected organs. As with other malignancies, tumor staging is also a well-known prognostic factor for HNC, but this study does not include the tumor stage. Thus, these factors would be a profound confounding factor in this study. However, many HNC patients have delayed symptom onset and are not diagnosed until the tumor has progressed to some extent. Therefore, the potential negative impact of the tumor stage not being reflected in the analysis is expected to be significantly lower than that of other malignant tumors. Second, this cohort dataset does not provide detailed information about what type of chemotherapy and radiotherapy each HNC patient received due to personal privacy policy in South Korea. Therefore, we were unable to analyze the effect of the type and duration of cancer treatment methods, such as chemotherapy and radiotherapy, on the risk of developing Alzheimer’s disease. This significant limitation must be overcome in future studies. Additionally, this cohort dataset does not include information on education level. However, some studies have linked low education to risk of developing Alzheimer’s disease, whereas another study indicated that higher educational attainment was positively correlated with faster cognitive decline [43,44,45]. Thus, future research with a detailed analysis according to education level is needed.

Third, according to previous studies, some lifestyle habits such as smoking and alcohol consumption are well known, not only as causes of HNC, but also as risk factors for Alzheimer’s disease. It is well known that a long history of smoking contributes to the higher development of HNC and a higher risk of second primary tumors than in the general population. Additionally, long-standing alcohol consumption aggravates the loss of brain cells and increases the number of amyloid plaques in the brain, which is a significant pathology of Alzheimer’s disease. However, in this study, we could not access personal information such as smoking and alcohol consumption habits; this issue is one of the critical limitations of the present study. Fourth, in recent years, it has become well known that human papillomavirus (HPV) infection is an important prognostic factor, especially for oropharyngeal cancer. Being HPV-positive showed excellent survival rates regardless of the treatment approach used. Apart from HPV, HNC patients with epidermal growth factor (EGFR) overexpression showed a poor prognosis. However, in this study, we did not perform a detailed analysis of HNC patients according to HPV and EGRF status. This has the potential to lead to selection bias. Key clinical test results that are thought to affect analysis results (fasting blood sugar, lipid profile, and liver enzyme levels) were also excluded. Fifth, the database used only the provided age as a group because of de-identification issues. Therefore, we had no choice but to match the two groups by classifying the age data rather than the actual age distribution, which may have reflected residual bias in our analysis. Finally, the fundamental limitations of our study design do not allow us to directly analyze the pathophysiological mechanisms between the two diseases. Therefore, studies that include a broader range of factors are needed to elucidate underlying pathophysiological mechanisms.

## 5. Conclusions

Our findings showed that HNC had a positive correlation with an increased development of Alzheimer’s disease. In particular, oral cavity cancer has shown a significantly higher risk of developing Alzheimer’s disease. For these reasons, we believe that the present study provides novel clinical insights for physicians regarding the relationship between HNC and Alzheimer’s disease. Therefore, we suggest that physicians should be aware of the potential development of Alzheimer’s disease in patients with HNC, and should monitor their cognitive decline.

## Figures and Tables

**Figure 1 cancers-15-05516-f001:**
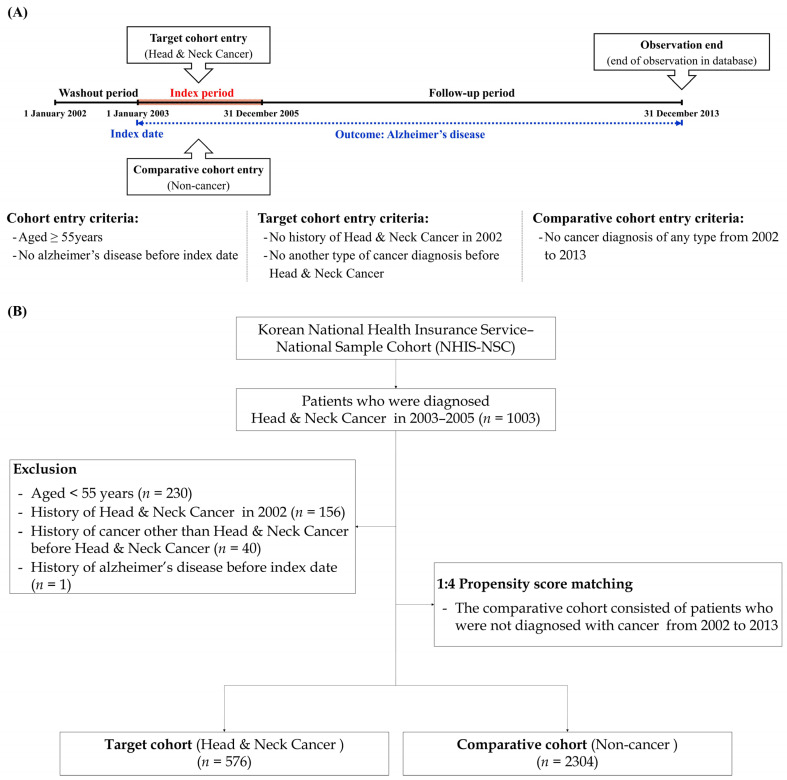
(**A**) Description of the study design; (**B**) Flow chart of study enrollment.

**Figure 2 cancers-15-05516-f002:**
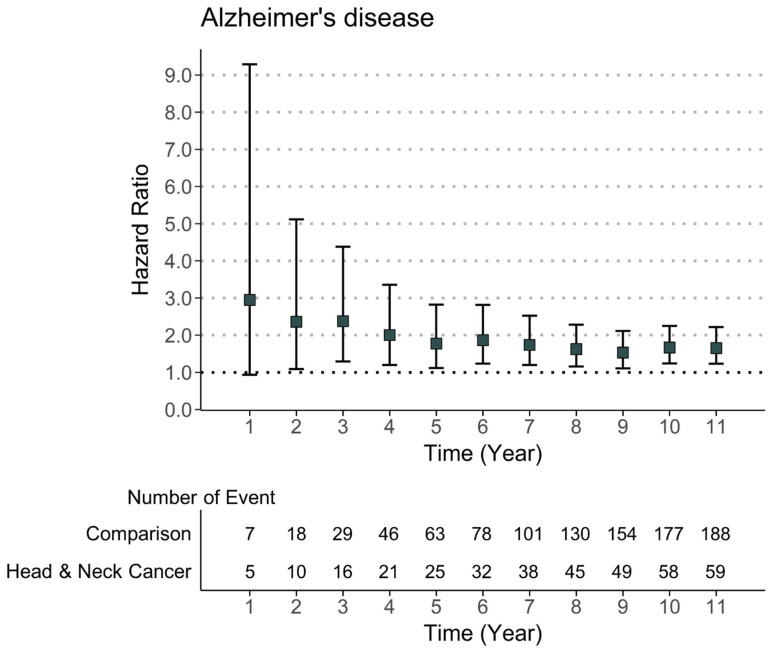
Risk of Alzheimer’s disease in patients with head and neck cancer by the follow-up time.

**Table 1 cancers-15-05516-t001:** Characteristics of the study participants.

Independent Variables	Comparative Cohort (Non-Cancer; *n* = 2304)	Target Cohort (Head and Neck Cancer; *n* = 576)	*p* Value
Sex			1.000
Male	1084 (47.0%)	271 (47.0%)	
Female	1220 (53.0%)	305 (53.0%)	
Age (years)			1.000
55–69	1276 (55.4%)	319 (55.4%)	
>69	1028 (44.6%)	257 (44.6%)	
Residence			1.000
Seoul	300 (13.0%)	75 (13.0%)	
Second area	368 (16.0%)	92 (16.0%)	
Third area	1636 (71.0%)	409 (71.0%)	
Household income			1.000
Low (0–30%)	616 (26.7%)	154 (26.7%)	
Middle (30–70%)	688 (29.9%)	172 (29.9%)	
High (70–100%)	1000 (43.4%)	250 (43.4%)	
CCI			1.000
0	1268 (55.0%)	317 (55.0%)	
1	604 (26.2%)	151 (26.2%)	
≥2	432 (18.8%)	108 (18.8%)	

Seoul, the largest metropolitan area; second area, other metropolitan cities; third area, other areas; CCI, Charlson Comorbidity Index.

**Table 2 cancers-15-05516-t002:** Comparison of incidence and risk rates of Alzheimer’s disease between HNC and non-cancer groups.

Group	N	Case	Person-Years	Incidence Rate	Crude HR (95% CI)	Weighted HR (95% CI)
Alzheimer’s disease						
Non-cancer	2304	188	19,249.3	9.77	1.00 (ref)	1.00 (ref)
HNC	576	59	3953.7	14.92	1.67 (1.24–2.24) ***	1.65 (1.23–2.22) ***

HNC; head and neck cancer, HR, hazard ratio; CI, confidence interval. *** *p* < 0.001.

**Table 3 cancers-15-05516-t003:** Hazard ratios of Alzheimer’s disease by sex between non-cancer and HNC groups.

Sex	Male		Female	
	Non-Cancer	HNC	Non-Cancer	HNC
Alzheimer’s disease				
Crude HR (95% CI)	1.00 (ref)	1.21 (0.70–2.07)	1.00 (ref)	1.92 (1.34–2.73) ***
Weighted HR (95% CI)	1.00 (ref)	1.21 (0.70–2.08)	1.00 (ref)	1.93 (1.35–2.75) ***

HNC; head and neck cancer, HR, hazard ratio; CI, confidence interval. *** *p* < 0.001.

**Table 4 cancers-15-05516-t004:** Hazard ratios of Alzheimer’s disease by age between comparison and HNC groups.

Age (Years)	55–69		>69	
	Non-Cancer	HNC	Non-Cancer	HNC
Alzheimer’s disease				
Crude HR (95% CI)	1.00 (ref)	2.30 (1.34–3.96) **	1.00 (ref)	1.50 (1.05–2.13) *
Weighted HR (95% CI)	1.00 (ref)	2.29 (1.33–3.93) **	1.00 (ref)	1.47 (1.03–2.10) *

HNC; head and neck cancer, HR, hazard ratio; CI, confidence interval. * *p* < 0.05, ** *p* < 0.010

**Table 5 cancers-15-05516-t005:** Incidence and risk of incident Alzheimer’s disease event according to the subtype of head and neck cancer.

Variables	N	Case	Person Year	Incidence Rate	Unadjusted HR (95% CI)	Adjusted HR (95% CI)
Cancer type						
Comparison	2304	188	19,249.3	9.77	1.00 (ref)	1.00 (ref)
Oral cavity	425	51	3075.2	16.58	1.84 (1.35–2.51) ***	
Salivary gland	8	0	45.3	-	0.00 (0-Inf)	0.00 (0-Inf)
Oropharynx	21	2	133.3	15.00	1.74 (0.43–7.01)	1.90 (0.47–7.69)
Nasopharynx	22	0	128.2	-	0.00 (0-Inf)	0.00 (0-Inf)
Hypopharynx	8	0	46.3	-	0.00 (0-Inf)	0.00 (0-Inf)
Sinonasal tract	7	0	24.1	-	0.00 (0-Inf)	0.00 (0-Inf)
Larynx	85	6	501.2	11.97	1.36 (0.60–3.06)	1.75 (0.77–4.01)

HR, hazard ratio; CI, confidence interval. *** *p* < 0.001.

## Data Availability

The datasets generated and/or analyzed in the current study are not publicly available due to the policy of the Korea National Health Insurance Service, but are available from the corresponding author upon reasonable request.

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
