# Peer review of "Increased Risk of Alzheimer’s Disease in Patients with Head and Neck Cancer"

_cancers, 2023, doi:10.3390/cancers15235516_

Round 1
Reviewer 1 Report
Comments and Suggestions for Authors
The manuscript entitled “Increased risk of Alzheimer’s disease in patients with head and neck cancer" by Lee et al is potentially an important study.
The following are some points that the authors should consider.
1. Authors should write more details of data collection method in materials and methods.
2. In Methodology, Authors should write in more details about the diagnostic code.
3. Include the references used in methodology.
4. As several factors like alcoholic consumption and other lifestyle factors that induce cancer as well as dementia. Did authors compare the dementia state between HNC patients with and without alcohol consumption?
5. Why did the author choose not to include the immigrant data?
6. In table 4 hazard ratio of Alzheimer disease by age between comparison and HNC groups. Authors should recheck the statistical analysis.
7. Review the manuscript for punctuation, spelling mistakes and long sentences
Thanks
Author Response
1. Authors should write more details of data collection method in materials and methods.
Answer: Thank you for your kind comment. As you recommended, we thoroughly added and modified the section on Method.
2. In Methodology, Authors should write in more details about the diagnostic code.
Answer: As you recommended, we added the diagnostic code in more detail as follows: We enrolled patients with HNC during the index period (2003–2005). The cancer group was defined based on the presence of diagnostic codes for HNC (oral cavity cancer: C00-C06, salivary cancer: C07-C08, oropharynx cancer: C09-C10, nasopharynx cancer: C11-C12, hypopharynx cancer: C13-C14, sinonasal cancer: C30-C31, and larynx cancer: C32)
3. Include the references used in methodology.
Answer: We added several references in the section of the Method.
4. As several factors like alcoholic consumption and other lifestyle factors that induce cancer as well as dementia. Did authors compare the dementia state between HNC patients with and without alcohol consumption?
Answer: We totally agree with your opinion. However, this cohort dataset does not provide alcoholic consumption and other lifestyle factors due to personal privacy policy issues in South Korea. Thus, we added this limitation in the section of the discussion as follows: "Third, according to previous studies, some lifestyle habits such as smoking and alcohol consumption are well known, not only as causes of HNC but also as risk factors for Alzheimer's disease. It is well-known that a long history of smoking contributed to the higher development of HNC and a higher risk of second primary tumors than the general population. Additionally, long-standing alcohol consumption aggravates the loss of brain cells and increases the number of amyloid plaques in the brain, which is a significant pathology of Alzheimer's disease. However, in this study, we could not access personal information such as smoking and high alcohol consumption habits, thus this issue is one of the critical limitations of the present study."
5. Why did the author choose not to include the immigrant data?
Answer: This is because the National Health Insurance Corporation in South Korea, which provides the cohort data set, judged that using a single ethnicity would reflect the Republic of Korea more representatively. Because South Korea has a very low proportion of immigrants from abroad.
6. In table 4 hazard ratio of Alzheimer disease by age between comparison and HNC groups. Authors should recheck the statistical analysis.
Answer: As the reviewer recommended, we rechecked the statistical analysis in Table 4 and we confirmed this data analysis.
7. Review the manuscript for punctuation, spelling mistakes and long sentences
Answer: The revised manuscript was corrected for English grammar and spelling by Editage.
Reviewer 2 Report
Comments and Suggestions for Authors
1. The present study identified that female patients with HNC showed a significantly increased risk of Alzheimer’s disease. Is the education level of the patients known? Recently, a systemic review with meta-analyses and other studies identified dementia and cognitive dysfunction in individuals with lower education levels, particularly women (PMIDs: 33313373; 35035379; 34814847). Please discuss.
2. Mechanism related to radiation, chemo and surgery induced changes causing possible dementia were nicely discussed. Is this study identified any link with specific chemo agent or a therapeutic module?
3. Since this a large-scale study is any specific comorbid condition identified as a risk factor for dementia?
Comments on the Quality of English LanguagePlease proofread for spelling and grammatical errors
Author Response
1. The present study identified that female patients with HNC showed a significantly increased risk of Alzheimer’s disease. Is the education level of the patients known? Recently, a systemic review with meta-analyses and other studies identified dementia and cognitive dysfunction in individuals with lower education levels, particularly women (PMIDs: 33313373; 35035379; 34814847). Please discuss.
Answer: Thank you for your kind comment. However, unfortunately, the cohort data set does not provide information related to education level. Thus, we described this issue in the section of the Discussion as follows: "Additionally, this cohort dataset does not include the information on education level. However, some studies have linked low education to the risk of developing Alzheimer's disease, whereas another study indicated that higher educational attainment was positively correlated with faster cognitive decline [43-45]. Thus, future research is needed with a detailed analysis according to education level."
2. Mechanism related to radiation, chemo and surgery induced changes causing possible dementia were nicely discussed. Is this study identified any link with specific chemo agent or a therapeutic module?
Answer: This cohort dataset does not provide detailed information about what type of chemotherapy and radiotherapy each HNC patient received due to personal privacy policy in South Korea. Therefore, we were unable to analyze the effect of the type and duration of cancer treatment methods, such as chemotherapy and radiotherapy, on the risk of developing Alzheimer's disease. This significant limitation must be overcome in future studies. Again, thank you for your kind comment.
3. Since this a large-scale study is any specific comorbid condition identified as a risk factor for dementia?
Answer: First of all, thank you for your careful review. In this study, we adjusted for other comorbidities between the two groups in order to analyze the risk of Alzheimer's disease solely due to the influence of HNC. Therefore, it is not possible to know the correlation between other comorbidities and Alzheimer's disease. In the future, we will continue research to uncover other diseases that are related to the risk of Alzheimer's disease.